# Objectively Measured Physical Activity and Sedentary Time of Suburban Toddlers Aged 12–36 Months

**DOI:** 10.3390/ijerph19116707

**Published:** 2022-05-31

**Authors:** Louise A. Kelly, Allan Knox, Carlos Gonzalez, Patrick Lennartz, Jordan Hildebrand, Blake Carney, Spencer Wendt, Rebecca Haas, Mason D. Hill

**Affiliations:** 1Exercise Science Department, California Lutheran University, Thousand Oaks, CA 93065, USA; aknox@callutheran.edu (A.K.); cgonzalez@callutheran.edu (C.G.); plennartz@callutheran.edu (P.L.); bcarney@callutheran.edu (B.C.); swendt@callutheran.edu (S.W.); haas@callutheran.edu (R.H.); mdhill@callutheran.edu (M.D.H.); 2Biology Department, California Lutheran University, Thousand Oaks, CA 93065, USA; jhildebrand@callutheran.edu

**Keywords:** pediatric, physical activity, sedentary behavior, ethnicity, tracking, Hispanic, African American

## Abstract

Low levels of physical activity may predispose children to the development of obesity and related chronic diseases in later life. The aims of this study were as follows: (1) quantitatively describe the levels of habitual physical activity in a contemporary sample of suburban children aged 12 to 36 months; (2) assess for gender differences in physical activity and sedentary behavior; (3) examine the specific effects of ethnicity, gender and overweight status on the objectively measured physical activity; and (4) quantify the tracking of physical activity in a subset of children over 1 year. During year one, 142 participants wore the GT3X Actigraph for 3 days. At a 1-year follow-up, a subset of 25 participants wore the Actigraph for 7 consecutive days. GLM and *t*-tests as appropriate were carried out to assess the influence of gender on the physical activity level. Spearman rank correlations, percentage agreement and kappa statistics assessed the tracking of physical activity. The results showed no significant gender differences in any anthropometric measurements, sedentary behavior or MVPA (*p* > 0.05). There were also no significant gender, ethnicity or overweight interaction for sedentary behavior, time spent in light PA and time spent in MVPA (*p* > 0.05). For tracking, there was a moderate strength of agreement for MVPA. Considering the disproportionate effects of obesity in minority groups, culturally appropriate interventions targeting the reduction in sedentary behavior are urgently required.

## 1. Introduction

Pediatric obesity is a very serious problem in the United States because of the adverse consequences in both the short and long term to children’s health [1,2]. Obese children are at high risk for diseases such as type 2 diabetes [3], cardiovascular disease [4] and obesity-related cancers [5]. It is currently estimated that 13.7 million children aged 2–19 years and 13.9% of children aged 2–5 are affected by obesity [6]. The prevalence of obesity is considerably higher among minority populations; in 2017, the Centres for Disease Control [7] publication reported that Hispanics and non-Hispanic Blacks had a higher obesity prevalence compared with non-Hispanic Whites and non-Hispanic Asians (25.8%, 22.0%, 14.1% and 11%, respectively). Furthermore, while participation in physical activity is low in the pediatric population, they are particularly low for Black and Hispanic children [8]. However, potential differences in physical activity and sedentary behavior levels of toddlers from different ethnic groups have yet to be established.

The dearth of research regarding physical activity and time spent in sedentary behaviors in toddlers represents a crucial gap in the literature. The toddler years are of particular importance, as it was established that the first 5 years of life are critical periods for the development of obesogenic behaviors [9,10,11,12]. Furthermore, a large proportion of children were shown to engage in high levels of screen time before the age of 3 years, thus significantly increasing their time spent being sedentary. Unfortunately, these physical activity and sedentary behaviors were shown to track from childhood to adulthood [13]. Data shows a change in energy intake and a decline in habitual physical activity and moderate-to-vigorous physical activity (MVPA), while sedentary behaviors dramatically increased [14,15,16,17]. While the health consequences of physical activity are well established for preschool-aged children [18], assessing physical activity in toddlers has not been a priority. This may be due to the belief that young children typically engage in short and sporadic bursts of activity and reactivity due to children’s inherently curious nature and are therefore healthy. However, evidence has demonstrated a relationship between adult-onset diseases originating in early childhood, placing urgency on the assessment of the physical activity of toddlers.

While a plethora of data on physical activity and sedentary behavior of preschool and older children exist in the literature, there is a paucity of data on toddlers. This may be due to theoretical challenges in measuring physical activity and sedentary behavior in toddlers [19]. Current recommendations suggest that 180 min of physical activity per day is needed for health in children aged 1–4 years [20]. However, Reilly et al. suggested that most young children are not meeting these recommendations, as most of their time is spent in sedentary behavior [1]. The levels of sedentary behavior may be more pronounced in minority communities but this is difficult to assess, as to our knowledge, there have been no data published on ethnic differences in sedentary behavior and different intensities of physical activity.

Tracking can be defined as the stability of a certain variable over time [21,22]. Tracking is traditionally assessed via test–test correlations demonstrating normative stability. The normal stability of a behavior is high when individuals remain at the same relative position within the sample distribution. Knowing one’s relative position and maintenance of that position within one’s cohort is a helpful tool in quantifying whether a child will maintain his/her relative rank for physical activity and sedentary behavior within a group of children over time [23]. Physical activity and sedentary behavior tracking studies not only quantify whether a child will maintain his or her relative rank for physical activity and sedentary behavior within a cohort of children over time but also are useful predictors of a child’s later physical activity and sedentary behavior based on their initial behavior [13,24]. Previous studies reported low levels of tracking of total physical activity, MVPA and sedentary behavior in children aged 3+ years [22,25]; no tracking data exists on children aged 12–36 months. Therefore, the aims of this study were as follows: (1) quantitatively describe the levels of habitual physical activity in a contemporary sample of suburban children aged 12 to 36 months; (2) assess for gender differences in physical activity and sedentary behavior; (3) examine the specific effects of ethnicity, gender and overweight status on the objectively measured physical activity; and (4) quantify the tracking of physical activity in a subset of children over 1 year.

## 2. Materials and Methods

### 2.1. Study Participants

A total of 142 children between the ages of 12 and 36 months and their parents were recruited to participate in this study. Participants and their families were recruited from local private, community daycare facilities, along with recruitment through word of mouth. For the present study, the children were required to provide a minimum of 3 complete days of physical activity. Therefore, a sample of 100 children (53 boys, 47 girls) was included in the final analyses. A subsample of 25 children provided 7 days of physical activity data at a 12-month follow-up. Parents were asked to report their child’s gender and ethnicity. For a child to be included in a particular ethnicity, the parents’ and grandparents’ descent had to be from the same ethnic group. The California Lutheran Universities Institutional Review Board approved the study (IRB Number 2011107) and informed written consent was obtained from the parent or guardian of each child. All methods were performed in accordance with the relevant guidelines and regulations.

### 2.2. Anthropometric Measures

Using a SECCA stadiometer and a Tanita medical scale and without shoes or socks, height was measured to the closest 0.1 cm and body weight to the closest 0.05 kg. Gender-specific weight and/or BMI percentiles for each child (where age-appropriate) were determined using EpiInfo, Version 7 for Mac (CDC, Atlanta, GA, USA).

### 2.3. Physical Activity and Sedentary Behavior

Habitual physical activity and sedentary behavior were assessed using the Actigraph GT3X accelerometer (Pensacola, FL, USA). The Actigraph GT3X was chosen based on the review by Cliff et al. [19]. The GT3X was securely attached to an elastic belt on the child’s right hip, as previously reported [26,27]. Children continuously wore the GT3X under clothing for a minimum of three consecutive days and a maximum of 7 consecutive days. A minimum of 6 h per day (i.e., 18 h over 3 days) of activity monitoring between the hours of 6:00 a.m. and 10:00 p.m. were required for inclusion in our final analysis. Parents were instructed to fasten the accelerometers on when the child woke up in the morning and remove it only for sleeping and bathing. Parents were also asked to log when the accelerometer was put on and removed. Data were collected in 15 s epochs. Non-wear time was defined as consecutive zero counts for ≥20 min and was, therefore, excluded from the analyses, as previously suggested by Vanderloo and Tucker [28]. To quantify the time spent in varying intensities of physical activity and in sedentary behavior, cut-off points from a previously published, peer-reviewed study were applied [29]. Sedentary activity (SED) was considered to be ≤181 counts/15 s epoch. Light physical activity (LPA) had a range of 182–434 counts/15 s epoch. Moderate-to-vigorous physical activity (MVPA) was ≥435 counts/15 s epoch. Children whose data did not meet the minimum time requirement of three days with six hours per day were not included in the final analysis.

### 2.4. Statistical Analysis

All data were checked for normality before the statistical analysis using descriptive statistics, histograms with normal distribution curves and Anderson–Darling (AD) normality tests. Student’s *t*-tests were carried out to assess the influence of gender on the physical activity level. General linear models (GLM) were used to examine the relationship between the independent variables (i.e., ethnicity, gender and overweight status) and the dependent variables (i.e., total physical activity, sedentary behaviors, light physical activity and MVPA). Analysis of variance (ANOVA) was used to examine the main and interaction effects of ethnicity, gender and overweight status. To assess the tracking, Spearman rank correlations were calculated to examine the tracking of physical activity and sedentary behavior between the baseline and the 12-month follow-up. Percentage agreement and kappa statistics were calculated between the baseline and 12-month follow-up to determine the likelihood that a particular child would be classified in the same group from year to year [30]. A kappa value of <0.20 represented a poor strength of agreement, 0.21 to 0.40 represented a fair strength of agreement, 0.41 to 0.60 represented a moderate strength of agreement, 0.61 to 0.80 represented a good strength of agreement and 0.81 to 1.00 represented a very good strength of agreement [30]. All analyses were conducted using SPSS (Mac version 24, IBM, Armonk, NY, USA) and an alpha of 0.05 was applied for all analyses.

## 3. Results

A total of 142 children aged 12–36 months were recruited for the study. The final analysis was conducted on 100 children; 42 children were not included for the following reasons: 30 failed to provide the minimum GT3X wear time and 12 refused the testing. At the baseline, the participants included 47 girls and 53 boys. The mean age at the baseline was 1.8 ± 0.5 months. At the baseline, *t*-tests showed no significant gender differences for age (*p* = 0.057), height (*p* = 0.07), weight (*p* = 0.34), BMI percentile (*p* = 0.51) or weight percentile (*p* = 0.32). The mean accelerometer measurement period was 8.03 waking hours per day (range 6.3–9.9 h) at the baseline. The baseline characteristics are shown in Table 1.

A subsample of children (*n* = 25; 10 girls and 15 boys) had their physical activity assessed 1 year later. The mean age at the 1-year follow-up was 2.74 ± 0.5 months. The *t*-tests showed no significant gender differences for age (*p* = 0.039), height (*p* = 0.63), weight (*p* = 0.61), BMI percentile (*p* = 0.42) or weight percentile (*p* = 0.51). The mean accelerometer measurement period was 9.12 waking hours per day (range 6.8–10.2 h). The 1-year follow-up characteristics are shown in Table 1.

At the baseline, the participants spent 56.24 ± 6.64% of time in SB, 1.8 ± 2.6% in LPA and 41.7 ± 7.0% in MVPA. The *t*-tests showed no significant gender difference for SB (57.27 ± 6.18 for girls vs. 55.4 ± 6.9 for boys; *p* = 0.807) or MVPA (40.77 ± 7.17 for girls vs. 42.44 ± 6.95 for boys; *p* = 0.714). However, there was a significant gender difference for LPA, with boys spending more time at this PA intensity than girls (2.12 ± 3.1 for boys vs. 1.40 ± 1.86 for girls; *p* = 0.031). When we examined the effects of ethnicity gender and overweight status on physical activity, the results from the GLM showed no significant gender, ethnicity or overweight interaction for sedentary behavior (*p* = 0.682), time spent in light PA (*p* = 0.15) and time spent in MVPA (both *p* = 0.64) (see Table 2).

The Spearman rank correlation and kappa statistics are presented in Table 3. The Spearman rank correlation for SB was r = 0.63 (*p* < 0.02) and kappa was 0.28 (*p* = 0.89), which showed a fair strength of agreement [30] The Spearman rank correlation for LPA was very low r = −0.003 (*p* = 0.99) and kappa was −0.018 (*p* = 0.56), which showed a poor strength of agreement. Meanwhile, the Spearman rank correlation for MVPA was r = 0.66 (*p* = 0.02) and kappa was 0.47 (*p* = 0.007), which showed a moderate strength of agreement see Table 3.

## 4. Discussion

It is well documented that the early years of development are critical years for the development of an active lifestyle and healthy living behaviors. Unfortunately, it is this exact period that we know the least about the impact of physical activity and the consequences of sedentary behavior. In 2019, the World Health Organization estimated that 38 million children under the age of 5 were overweight or obese. Declines in physical activity and increases in sedentary behavior have significantly contributed to the epidemic of obesity in childhood [14,15]. Therefore, the health implications of physical activity during the early years cannot be disregarded. Specifically, targeting physical activity and sedentary behavior is undoubtedly the only evidence-based approach for obesity prevention.

To our knowledge, this is the first study to assess ethnic differences in habitual physical activity and sedentary behavior and quantify tracking in a suburban representative sample of toddlers. The key findings from our study showed that toddlers, regardless of their gender or ethnic identity, spent a considerable amount of their waking hours engaged in sedentary behaviors. This study suggested that gender differences in sedentary behavior and MVPA commonly seen in children from preschool age and onward were not observed at this very young age. However, we did observe gender differences regarding boys spending more time in light PA compared with girls. Our findings that children were spending a large amount of their waking time in sedentary behavior are consistent with a growing body of evidence [1,31].

However, a direct comparison of our data with other published data is rather difficult given the array of tools to assess physical activity and sedentary behavior in this age group. A study by Carson and Kuzik found somewhat similar results to our study. Their study comprising urban Canadian toddlers found that females, those from ethnic minority groups and toddlers from families of lower socio-economic groups were significantly more sedentary by engaging in more video and computer use [32]. A review by Prioreschi and colleagues reported on the physical activity levels of children under 2 years old; however, due to the vast array of methods used to assess physical activity, synthesis of the results is impossible. While only six of the studies used accelerometers, none of the studies reported on sedentary behavior or tracking of these behaviors [33].

A review and meta-analysis of 47 studies showed that children spent a similar amount of time in sedentary behavior to those in this current study. However, there were some limitations to this analysis of the Pereira review, namely, the accelerometer wear time criterion was not applied; therefore, there is a possibility that the sedentary behavior data used may not accurately assess daily habitual levels [34]. Other studies have also reported similarly high levels of sedentary behavior to those found in the current study. However, we report higher levels of time spent in MVPA and less time in light physical activity. This may, in part, have been due to our sample population being a suburban one and conducted in a warmer climate, as the weather was shown to affect participation in physical activity [35]. This study also suggested that gender differences in engagement in light physical activity were similar to those found in older children. However, these differences should be viewed with caution. A mean difference of 0.72 between the sexes in accelerometer output should not be interpreted as a 0.72% difference in light physical activity.

Of foremost concern are the ever-widening ethnic inequalities on obesity prevalence. Differences in physical activity (PA) may contribute to this risk but few data exist in childhood, and to our knowledge, no studies exist in toddlers. There are blatant differences in childhood adiposity and metabolic disorders within certain ethnic minority groups [36,37]. These differences may be associated with existing disparities, such as socioeconomic and ethnic differences. Literature in adult and older children’s population show ethnicity to be a crucial predictor of obesity and metabolic health. Studies found that, in general, non-White children are less physically active and more sedentary than their White peers [38]. While the results of our study found no significant gender, ethnicity and obesity interaction, it should be noted that young Asian females spent the most time in SED, African American boys spent the most time in LPA and Hispanic boys spent the most time in MVPA and the lowest time in SB. Our results would suggest that ethnic differences in physical activity and sedentary behaviors are established much earlier than anticipated, leading to the need to intervene at a much earlier age than originally thought.

To our knowledge, this is the first study to report the tracking of physical activity and sedentary behavior in suburban toddlers. The results of our study showed that the tracking coefficients between the baseline and 12-month follow-up showed moderate levels of tracking for sedentary behavior and MVPA and a negligible correlation for LPA. The kappa statistics ranged from −0.02 to 0.47, demonstrating the maintenance of MVPA and sedentary behavior but not LPA in toddlers. These results again emphasized the need for successful interventions grounded in the comprehension of the very intricate nature of these behaviors in this age group. While there is a very small body of evidence of physical activity behaviors in toddlers, to our knowledge, there are no currently published studies reporting on the objectively measured tracking of sedentary behavior or physical activity behaviors in this young age group. There are published data on tracking in preschool-aged children; however, only a few studies have used accelerometers and others have used a range of methods, such as questionnaires in a variety of cultures and age groups [24,39,40,41]. Jackson and colleagues reported similar levels of tracking to ours in a sample of preschool-aged children using an older model of the Actigraph [39]. In contrast, other studies found lower levels of tracking in children aged 3–5 years by using Actigraph, heart rate and direct observations [24,42,43].

It is important to note that the present study has several limitations. First, because we used three consecutive days and six waking hours as our minimum criteria for measurement, caution is required if trying to compare our results to a seven-day monitoring period. Second, the number of participants in our tracking study was relatively low. While tracking studies in older age groups had larger sample sizes, it was difficult for us to assess the sample size needed for this study, as no other tracking studies in this age group have been published. However, given that our group was tracked over 12 months using objective measurements, age-appropriate cut-offs and robust statistical techniques to assess tracking, we are confident that our report of moderate levels of tracking in this age group is correct. Lastly, the numbers of Asian males and African American girls recruited for the study were low.

Several strengths should also be noted. Our study was the first relatively large, ethnically diverse and socioeconomically representative sample of very young suburban children. As noted by Cliff et al., methodological studies at this young age group are lacking; therefore, protocols must be extrapolated from studies of preschool-aged children. Based on this, we used the objective measurement of accelerometers and, in particular, the Actigraph, with a minimum wear period of three consecutive days for 6 h of “waking time”, which is consistent with studies in the preschool age [1,11,13,19,26]. Furthermore, several recent studies in laboratory and field settings have examined feasibility and placement issues in toddlers, where the results have shown that use of the accelerometers is a valid and reliable tool for the assessment of physical activity and sedentary behaviors in this young age group [44,45,46]. However, there is still no consensus on placement, with studies recommending placement on the hip [45,46] and others recommending ankle placement [47]. However, Kwon et al. suggested that hip placement of an Actigraph is more feasible than wrist placement in this age group [45]. The precision of our assessment of physical activity and sedentary behavior in this young group is exemplified by the use of age-appropriate cut-off points with 15 s epochs; defining a non-wear time as 20 min of consecutive zero’s, thus capturing a more accurate picture of this age groups patterns; and finally, the multiple indices to capture the tracking. In addition to the traditional use of correlations, we calculated kappa statistics; by calculating these, we were able to reaffirm the extent to which a participant remained in their respective physical activity and sedentary behavior categories over the 12 months.

## 5. Conclusions

In summary, this was the first study to objectively measure habitual physical activity and sedentary behavior, quantify tracking and assess ethnic differences in habitual physical activity and sedentary behavior in a suburban representative sample of toddlers. The results of this study raise the possibility that interventions targeting toddlers, especially minority children, should focus on understanding the ethnicity disparities in physical activity and sedentary behavior and then focus on promoting culturally appropriate interventions to increase physical activity, decrease sedentary behavior and, in turn, change the trajectory of these negative patterns before they become lifelong. Practical implications may include designing culturally appropriate health literacy interventions to teach parents about the importance of lifelong physical activity and the physiological consequences of a sedentary lifestyle on their young child.

## Figures and Tables

**Table 1 ijerph-19-06707-t001:** Participants characteristics.

	Cross-Sectional Study (*n* = 100)		1-Year Follow-Up Study (*n* = 25)	
	Girls (*n* = 47)	Boys (*n* = 53)	Girls (*n* = 10)	Boys (*n* = 15)
Age (years)	1.81 ± 0.49	1.89 ± 0.59	3.0 ± 0.47	2.54 ± 0.52
Height (m)	0.84 ± 0.16	0.89 ± 0.07	1.00 ± 0.44	0.94 ± 0.53
Weight (kg)	13.06 ± 1.93	13.45 ± 1.67	16.14 ± 2.91	15.49 ± 1.57
BMI (kg/m^2^)	17.45 ± 2.83	16.56 ± 1.00	16.03 ± 1.98	17.24 ± 1.87
Weight Percentile (%)	57.53 ± 30.21	64.35 ± 20.87	64.22 ± 25.83	73.00 ± 23.76
BMI Percentile (%)	61.73 ± 33.48	56.23 ± 25.42	55.12 ± 36.83	70.17 ± 32.22

Note: all data are given as mean ± standard deviation.

**Table 2 ijerph-19-06707-t002:** Participant characteristics by ethnicity.

	White Girls (*n* = 19)	White Boys (*n* = 19)	Hispanic Girls (*n* = 15)	Hispanic Boys (*n* = 18)	Asian Girls (*n* = 7)	African American Girls (*n* = 6)	African American Boys (*n* = 16)
Age (yrs)	1.7 ± 0.5	1.8 ± 0.5	1.85 ± 0.41	2.1 ± 0.7	1.9 ± 0.6	1.5 ± 0.7	2.0 ± 0.5
Height (m)	0.88 ± 0.08	0.89 ± 0.6	0.83 ± 0.2	0.91 ± 0.01	0.85 ± 0.1	0.68 ± 0.4	0.88 ± 0.1
Weight (kg)	12.9 ± 2.2	13.3 ± 1.6	13.3 ± 2.2	14.1 ± 2.1	12.2 ± 0.8	14.1 ± 0.5	13.2 ± 1.4 *
BMI percentile (%)	45.4 ± 36.4	51.6 ± 23.3	73.1 ± 34.3	59.7 ± 31.8	49.0 ± 15.7	84.4 ± 17.4	64.5 ± 20.6
Weight percentile (%)	49.1 ± 31.0	60.8 ± 18.4	59.8 ± 4.4	67.7 ± 29.4	51.9 ± 23.9	94.9 ± 1.7	65.0 ± 6.1
SB (%)	54.2 ± 6.5	56.9 ± 7.6	59.7 ± 4.4	53.8 ± 7.5 *	60.3 ± 5.7	56.6 ± 7.0	54.9 ± 6.2
LPA (%)	1.9 ± 2.8	1.1 ± 0.2	1.0 ± 0.4	2.2 ± 3.3	1.0 ± 0.1	1.2 ± 0.1	3.2 ± 4.2
MVPA(%)	43.9 ± 7.3	41.9 ± 7.6	37.6 ± 6.2	44.0 ± 7.9 *	38.5 ± 5.6	42.2 ± 7.1	41.8 ± 6.1

Note: all data are given as mean ± standard deviation; * denotes significance; SB (%) denotes the percentage of time spent in sedentary behavior, LPA (%) denotes the percentage of time spent in light physical activity and MVPA denotes the percentage of time spent in moderate-to-vigorous physical activity.

**Table 3 ijerph-19-06707-t003:** Spearman rank-order correlations and kappa statistics to assess tracking over 1 year.

	Spearman Rank Correlation		Percentage Agreement (%)	Kappa	Strength
	R-Value	*p*-Value			
SB (%)	0.626	0.02 *	28.6	0.89	Fair strength
LPA (%)	−0.003	0.99	1.8	0.56	Poor agreement
MVPA (%)	0.662	0.02 *	46.9	0.007 *	Moderate strength

Note: * denotes significance.

## Data Availability

Data is available upon written request to the corresponding author.

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
