# Peer review of "Objectively Measured Physical Activity and Sedentary Time of Suburban Toddlers Aged 12–36 Months"

_ijerph, 2022, doi:10.3390/ijerph19116707_

Round 1

Reviewer 1 Report

The paper is interesting because it raises a problem that has not been addressed by the scientific literature.

however major revisions are required for publication.

First of all, the sampling number is low, some cells have a very low number of observations (Asian boys n = 2; African American boys n = 3) which prevent any statistical evaluation. Although the authors themselves highlight this limitation, then they improperly present statistical data in table 2, which must therefore be eliminated (i.e. they indicate a trend for significance for the variable Age in Asian boys group, which consists of 2 subjects).

Of note, subjects included in table 2 seem not to correspond to subjects in table 1. In table 1 there are 47 girls and 53 boys. In table 2 there are 44 girls and 56 boys. Please clarify.

Another very important aspect is the References, which are often very dated. Many statements, especially in the introduction, are supported by very dated references (1996-1999).

Finally, English needs to be revised.

Author Response

Reviewer 1 Comments

Dear Reviewer 1, thank you for taking the time to read and review our paper. We hope we have addressed your concerns adequately.

Dr. Louise Kelly

Reviewer Comment: First of all, the sampling number is low, some cells have a very low number of observations (Asian boys n = 2; African American boys n = 3) which prevent any statistical evaluation. Although the authors themselves highlight this limitation, then they improperly present statistical data in table 2, which must therefore be eliminated (i.e. they indicate a trend for significance for the variable Age in Asian boys group, which consists of 2 subjects).

Our response: Eliminated as requested

Reviewer Comment: Of note, subjects included in table 2 seem not to correspond to subjects in table 1. In table 1 there are 47 girls and 53 boys. In table 2 there are 44 girls and 56 boys. Please clarify.

Our response: Thank you, we have now corrected the numbers of participants in our tables.

Reviewer Comment: Another very important aspect is the References, which are often very dated. Many statements, especially in the introduction, are supported by very dated references (1996-1999).

 Our response: Updated as requested. However, some of these are imperative supporting reference so are a research group strongly felt we needed to include some “older” reference.

Reviewer Comment: Finally, English needs to be revised.

Our response: Revised as requested.

Reviewer 2 Report

General comments

The authors have clearly stated that the purpose of the study was assess differences in sedentary behavior and physical activity by gender; examine the interaction of ethnicity, gender, and overweight status on physical activity and sedentary behavior, and finally track physical activity in a subset of children aged 12-36 months over 1 year. The paper is well-written and easy to follow. In my opinion, it adds considerable value to the current literature, since physical inactivity has been considered as the greatest threat to public health globally. This study can enhance future attempts in similar research area in order to investigate more specific pathways between physical activity levels and health status in various populations. However, I have highlighted a few suggestions and concerns in my specific comments section (below) that need to be addressed before considering whether this work should be published or not.

Specific comments

ABSTRACT

  • Add a few sentences about type 1 diabetes and physical activity at the beginning.

  1. INTRODUCTION & DISCUSSION

  • Nice work from the authors. I suggest adding a statement aiming to connect the topic with the current state of the physical activity and exercise sector, and more specifically with the top relevant trends in this sector such as exercise for weight loss, outdoor activities and others, which appear to be very attractive in the USA and worldwide. Thus, consider citing the following study.

Kercher VM, Kercher K, Bennion T, Levy P, Alexander C, Amaral PC, et al. 2022 Fitness Trends from Around the Globe. ACSMs Health Fit J 2022; 26(1): 21–37.

  1. REFERENCES

  • Please delete journal instructions for authors regarding the references.

  1. CONCLUSIONS

  • Please add a separate paragraph to present more practical implications.

Author Response

Reviewer 2

Dear Reviewer 2, thank you for taking the time to read and review our paper. We hope we have addressed your concerns adequately.

Dr. Louise Kelly

 Reviewer Comments

ABSTRACT

  • Add a few sentences about type 1 diabetes and physical activity at the beginning.

Our response: Completed as requested

  1. INTRODUCTION & DISCUSSION

  • Nice work from the authors. I suggest adding a statement aiming to connect the topic with the current state of the physical activity and exercise sector, and more specifically with the top relevant trends in this sector such as exercise for weight loss, outdoor activities and others, which appear to be very attractive in the USA and worldwide. Thus, consider citing the following study.

Kercher VM, Kercher K, Bennion T, Levy P, Alexander C, Amaral PC, et al. 2022 Fitness Trends from Around the Globe. ACSMs Health Fit J 2022; 26(1): 21–37.

Our response: Having read the article above, we decided it would not significantly add to our manuscript so we did not include.

  1. REFERENCES
  • Please delete journal instructions for authors regarding the references.

Our response: Completed as requested.

  1. CONCLUSIONS
  • Please add a separate paragraph to present more practical implications.

Our response: Completed as requested.

Reviewer 3 Report

I appreciate your work. I made a number of technical type comments, and a few about what you wrote. I believe cleaning up your manuscript is some work, but the words and table information all seem appropriate. Your work is important to get out to the world.

  • Abstract, does not matter to me, but it seems MDPI does not use sections in the abstract like Abstract: METHODS, etc.
  • Line 11, no need for finally. Just a thought.
  • Line 13, unsure whether t-test needs to be t-test; same with p
  • Line 19, you wrote (p.0.05). I am not sure what that means.
  • Line 19-20, what greater understanding is needed? A specific conclusion seems better than your general statement.
  • The reference system for MDPI is [ref#]. Please make the change throughout.
  • Line 69, seems like an extra space between the 23 and Physical
  • Introduction seems well-written and referenced. There is a plethora of data in children, but not much in toddlers.
  • Line 75-80, You have four distinct purposes, but I cannot tell if they line up with your abstract. What is wrong with the same list in the abstract?
  • Line 92, perhaps include the IRB#.
  • Line 100, not sure you need italics
  • Line 123, seems an extra space Student’ s
  • Line 126, why is total physical activity (cpm), what is the cpm?
  • Line 133, and anywhere else, 0.xx; you seemed to switch to .xx
  • Line 145, Table 1 is just Table 1.
  • Line 155 is an example of space = and then other times is it no space=
  • Table 2, I see no reason for you’re the underlining
  • Again, all the 0.xx, .xx, 0.x, etc. Too confusing as to your style or reasoning
  • Line 165, why not just write standard deviations
  • Table 3, here you start using italics R value P value
  • You have SB( and then LPA space (
  • The inconsistencies make reading the words difficult. However, they are good words.
  • Conclusion, what are example disparity factors? Maybe provide a few examples.
  • Line 300, I see the IRB number, then the ) seems off.

Author Response

Reviewer 3

 Dear Reviewer 3, thank you for taking the time to read and review our paper. We hope we have addressed your concerns adequately.  We believe we have incorporated al of your suggestions listed below into our paper.

Dr. Kelly

  • Abstract, does not matter to me, but it seems MDPI does not use sections in the abstract like Abstract: METHODS, etc.  – removed section names
  • Line 11, no need for finally. Just a thought. - removed
  • Line 13, unsure whether t-test needs to be t-test; same with p - amended
  • Line 19, you wrote (p.0.05). I am not sure what that means. – Corrected to read > 0.05
  • Line 19-20, what greater understanding is needed? A specific conclusion seems better than your general statement. – We added a new conclusion
  • The reference system for MDPI is [ref#]. Please make the change throughout.  – Changed
  • Line 69, seems like an extra space between the 23 and Physical – Changed
  • Introduction seems well-written and referenced. There is a plethora of data in children, but not much in toddlers.  – Not sure what you were asking, can you please clarify
  • Line 75-80, You have four distinct purposes, but I cannot tell if they line up with your abstract. What is wrong with the same list in the abstract? – Changed
  • Line 92, perhaps include the IRB#. - added
  • Line 100, not sure you need italics - removed
  • Line 123, seems an extra space Student’ s  - removed
  • Line 126, why is total physical activity (cpm), what is the cpm? – sorry removed
  • Line 133, and anywhere else, 0.xx; you seemed to switch to .xx. – corrected
  • Line 145, Table 1 is just Table 1. – Changed
  • Line 155 is an example of space = and then other times is it no space= – Changed
  • Table 2, I see no reason for you’re the underlining - removed
  • Again, all the 0.xx, .xx, 0.x, etc. Too confusing as to your style or reasoning – Changed
  • Line 165, why not just write standard deviations – Changed
  • Table 3, here you start using italics R value P value – Changed
  • You have SB( and then LPA space ( – Changed
  • The inconsistencies make reading the words difficult. However, they are good words. Apologies
  • Conclusion, what are example disparity factors? Maybe provide a few examples. – Changed
  • Line 300, I see the IRB number, then the ) seems off. – Changed

Round 2

Reviewer 1 Report

The paper can be accepted in the current form